# Train longer, generalize better: closing the generalization gap in large batch training of neural networks

**Elad Hoffer**,[*]    **Itay Hubara**,[*]    **Daniel Soudry**

Technion - Israel Institute of Technology, Haifa, Israel

{elad.hoffer, itayhubara, daniel.soudry}@gmail.com

## Abstract

**Background:** Deep learning models are typically trained using stochastic gradient descent or one of its variants. These methods update the weights using their gradient, estimated from a small fraction of the training data. It has been observed that when using large batch sizes there is a persistent degradation in generalization performance - known as the "generalization gap" phenomenon. Identifying the origin of this gap and closing it had remained an open problem.

**Contributions:** We examine the initial high learning rate training phase. We find that the weight distance from its initialization grows logarithmically with the number of weight updates. We therefore propose a "random walk on a random landscape" statistical model which is known to exhibit similar "ultra-slow" diffusion behavior. Following this hypothesis we conducted experiments to show empirically that the "generalization gap" stems from the relatively small number of updates rather than the batch size, and can be completely eliminated by adapting the training regime used. We further investigate different techniques to train models in the large-batch regime and present a novel algorithm named "Ghost Batch Normalization" which enables significant decrease in the generalization gap without increasing the number of updates. To validate our findings we conduct several additional experiments on MNIST, CIFAR-10, CIFAR-100 and ImageNet. Finally, we reassess common practices and beliefs concerning training of deep models and suggest they may not be optimal to achieve good generalization.

## 1 Introduction

For quite a few years, deep neural networks (DNNs) have persistently enabled significant improvements in many application domains, such as object recognition from images (He et al., 2016); speech recognition (Amodei et al., 2015); natural language processing (Luong et al., 2015) and computer games control using reinforcement learning (Silver et al., 2016; Mnih et al., 2015).

The optimization method of choice for training highly complex and non-convex DNNs, is typically stochastic gradient decent (SGD) or some variant of it. Since SGD, at best, finds a local minimum of the non-convex objective function, substantial research efforts are invested to explain DNNs ground breaking results. It has been argued that saddle-points can be avoided (Ge et al., 2015) and that "bad" local minima in the training error vanish exponentially (Dauphin et al., 2014; Choromanska et al., 2015; Soudry & Hoffer, 2017). However, it is still unclear why these complex models tend to generalize well to unseen data despite being heavily over-parameterized (Zhang et al., 2017).

A specific aspect of generalization has recently attracted much interest. Keskar et al. (2017) focused on a long observed phenomenon (LeCun et al., 1998a) – that when a large batch size is used while

---

[*]Equal contribution

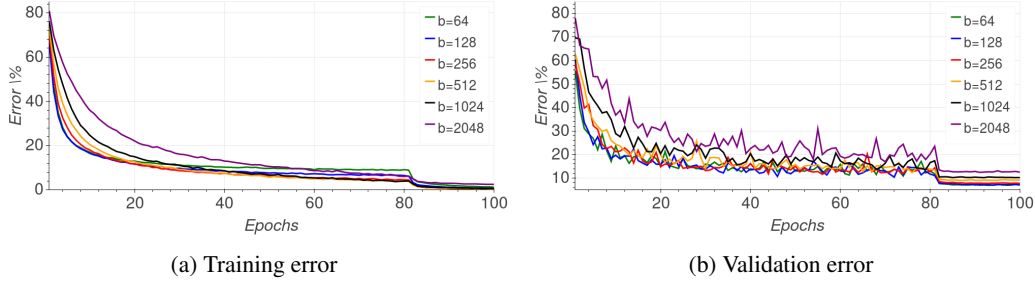

(a) Training error          (b) Validation error

Figure 1: Impact of batch size on classification error

training DNNs, the trained models appear to generalize less well. This remained true even when the models were trained "without any budget or limits, until the loss function ceased to improve" (Keskar et al., 2017). This decrease in performance has been named the "generalization gap".

Understanding the origin of the generalization gap, and moreover, finding ways to decrease it, may have a significant practical importance. Training with large batch size immediately increases parallelization, thus has the potential to decrease learning time. Many efforts have been made to parallelize SGD for Deep Learning (Dean et al., 2012; Das et al., 2016; Zhang et al., 2015), yet the speed-ups and scale-out are still limited by the batch size.

In this study we suggest a first attempt to tackle this issue.
First,

- We propose that the initial learning phase can be described using a high-dimensional "random walk on a random potential" process, with an "ultra-slow" logarithmic increase in the distance of the weights from their initialization, as we observe empirically.

Inspired by this hypothesis, we find that

- By simply adjusting the learning rate and batch normalization the generalization gap can be significantly decreased (for example, from $5\%$ to $1\% - 2\%$).

- In contrast to common practices (Montavon et al., 2012) and theoretical recommendations (Hardt et al., 2016), generalization keeps improving for a long time at the initial high learning rate, even without any observable changes in training or validation errors. However, this improvement seems to be related to the distance of the weights from their initialization.

- There is no inherent "generalization gap": large-batch training can generalize as well as small batch training by adapting the number of iterations.

## 2 Training with a large batch

**Training method.** A common practice of training deep neural networks is to follow an optimization "regime" in which the objective is minimized using gradient steps with a fixed learning rate and a momentum term (Sutskever et al., 2013). The learning rate is annealed over time, usually with an exponential decrease every few epochs of training data. An alternative to this regime is to use an adaptive per-parameter learning method such as Adam (Kingma & Ba, 2014), Rmsprop (Dauphin et al.) or Adagrad (Duchi et al., 2011). These methods are known to benefit the convergence rate of SGD based optimization. Yet, many current studies still use simple variants of SGD (Ruder, 2016) for all or part of the optimization process (Wu et al., 2016), due to the tendency of these methods to converge to a lower test error and better generalization.

Thus, we focused on momentum SGD, with a fixed learning rate that decreases exponentially every few epochs, similarly to the regime employed by He et al. (2016). The convergence of SGD is also known to be affected by the batch size (Li et al., 2014), but in this work we will focus on generalization. Most of our results were conducted on the Resnet44 topology, introduced by He et al. (2016). We strengthen our findings with additional empirical results in section 6.

**Empirical observations of previous work.** Previous work by Keskar et al. (2017) studied the performance and properties of models which were trained with relatively large batches and reported the following observations:

- Training models with large batch size increase the generalization error (see Figure 1).
- This "generalization gap" seemed to remain even when the models were trained without limits, until the loss function ceased to improve.
- Low generalization was correlated with "sharp" minima[2] (strong positive curvature), while good generalization was correlated with "flat" minima (weak positive curvature).
- Small-batch regimes were briefly noted to produce weights that are farther away from the initial point, in comparison with the weights produced in a large-batch regime.

Their hypothesis was that a large estimation noise (originated by the use of mini-batch rather than full batch) in small mini-batches encourages the weights to exit out of the basins of attraction of sharp minima, and towards flatter minima which have better generalization. In the next section we provide an analysis that suggest a somewhat different explanation.

## 3   Theoretical analysis

**Notation.** In this paper we examine Stochastic Gradient Descent (SGD) based training of a Deep Neural Network (DNN). The DNN is trained on a finite training set of $N$ samples. We define $\mathbf{w}$ as the vector of the neural network parameters, and $L_n(\mathbf{w})$ as loss function on sample $n$. We find $\mathbf{w}$ by minimizing the training loss.

$$L(\mathbf{w}) \triangleq \frac{1}{N} \sum_{n=1}^{N} L_n(\mathbf{w}) \,,$$

using SGD. Minimizing $L(\mathbf{w})$ requires an estimate of the gradient of the negative loss.

$$\mathbf{g} \triangleq \frac{1}{N} \sum_{n=1}^{N} \mathbf{g}_n \triangleq -\frac{1}{N} \sum_{n=1}^{N} \nabla L_n(\mathbf{w})$$

where $\mathbf{g}$ is the true gradient, and $\mathbf{g}_n$ is the per-sample gradient. During training we increment the parameter vector $\mathbf{w}$ using only the mean gradient $\hat{\mathbf{g}}$ computed on some mini-batch $B$ – a set of $M$ randomly selected sample indices.

$$\hat{\mathbf{g}} \triangleq \frac{1}{M} \sum_{n \in B} \mathbf{g}_n \,.$$

In order to gain a better insight into the optimization process and the empirical results, we first examine simple SGD training, in which the weights at update step $t$ are incremented according to the mini-batch gradient $\Delta \mathbf{w}_t = \eta \hat{\mathbf{g}}_t$. With respect to the randomness of SGD,

$$\mathbb{E}\hat{\mathbf{g}}_t = \mathbf{g} = -\nabla L(\mathbf{w}_t) \,,$$

and the increments are uncorrelated between different mini-batches[3]. For physical intuition, one can think of the weight vector $\mathbf{w}_t$ as a particle performing a random walk on the loss ("potential") landscape $L(\mathbf{w}_t)$. Thus, for example, adding momentum term to the increment is similar to adding inertia to the particle.

**Motivation.** In complex systems (such as DNNs) where we do not know the exact shape of the loss, statistical physics models commonly assume a simpler description of the potential as a random process. For example, Dauphin et al. (2014) explained the observation that local minima tend to have

low error using an analogy between $L(\mathbf{w})$, the DNN loss surface, and the high-dimensional Gaussian random field analyzed in Bray & Dean (2007), which has zero mean and auto-covariance

$$\mathbb{E}\left(L\left(\mathbf{w}_1\right) L\left(\mathbf{w}_2\right)\right) = f\left(\|\mathbf{w}_1 - \mathbf{w}_2\|^2\right) \tag{1}$$

for some function $f$, where the expectation now is over the randomness of the loss. This analogy resulted with the hypothesis that in DNNs, local minima with high loss are indeed exponentially vanishing, as in Bray & Dean (2007). Only recently, similar results are starting to be proved for realistic neural network models (Soudry & Hoffer, 2017). Thus, a similar statistical model of the loss might also give useful insights for our empirical observations.

**Model: Random walk on a random potential.** Fortunately, the high dimensional case of a particle doing a "random walk on a random potential" was extensively investigated already decades ago (Bouchaud & Georges, 1990). The main result of that investigation was that the asymptotic behavior of the auto-covariance of a random potential[4],

$$\mathbb{E}\left(L\left(\mathbf{w}_1\right) L\left(\mathbf{w}_2\right)\right) \sim \|\mathbf{w}_1 - \mathbf{w}_2\|^\alpha \ , \ \alpha > 0 \tag{2}$$

in a certain range, determines the asymptotic behavior of the random walker in that range:

$$\mathbb{E}\|\mathbf{w}_t - \mathbf{w}_0\|^2 \sim (\log t)^{\frac{4}{\alpha}} \ . \tag{3}$$

This is called an "ultra-slow diffusion" in which, typically $\|\mathbf{w}_t - \mathbf{w}_0\| \sim (\log t)^{2/\alpha}$, in contrast to standard diffusion (on a flat potential), in which we have $\|\mathbf{w}_t - \mathbf{w}_0\| \sim \sqrt{t}$. The informal reason for this behavior (for any $\alpha > 0$), is that for a particle to move a distance $d$, it has to pass potential barriers of height $\sim d^{\alpha/2}$, from eq. (2). Then, to climb (or go around) each barrier takes exponentially long time in the height of the barrier: $t \sim \exp(d^{\alpha/2})$. Inverting this relation, we get eq. $d \sim (\log(t))^{2/\alpha}$. In the high-dimensional case, this type of behavior was first shown numerically and explained heuristically by Marinari et al. (1983), then rigorously proven for the case of a discrete lattice by Durrett (1986), and explained in the continuous case by Bouchaud & Comtet (1987).

### 3.1 Comparison with empirical results and implications

To examine this prediction of ultra slow diffusion and find the value of $\alpha$, in Figure 2a, we examine $\|\mathbf{w}_t - \mathbf{w}_0\|$ during the initial training phase over the experiment shown in Figure 1. We found that the weight distance from initialization point increases logarithmically with the number of training iterations (weight updates), which matches our model with $\alpha = 2$:

$$\|\mathbf{w}_t - \mathbf{w}_0\| \sim \log t \ . \tag{4}$$

Interestingly, the value of $\alpha = 2$ matches the statistics of the loss estimated in appendix section B.

Moreover, in Figure 2a, we find that a very similar logarithmic graph is observed for all batch sizes. Yet, there are two main differences. First, each graph seems to have a somewhat different slope (i.e., it is multiplied by different positive constant), which peaks at $M = 128$ and then decreases with the mini-batch size. This indicates a somewhat different diffusion rate for different batch sizes. Second, since we trained all models for a constant number of epochs, smaller batch sizes entail more training iterations in total. Thus, there is a significant difference in the number of iterations and the corresponding weight distance reached at the end of the initial learning phase.

This leads to the following informal argument (which assumes flat minima are indeed important for generalization). During the initial training phase, to reach a minima of "width" $d$ the weight vector $\mathbf{w}_t$ has to travel at least a distance $d$, and this takes a long time – about $\exp(d)$ iterations. Thus, to reach wide ("flat") minima we need to have the highest possible diffusion rates (which do not result in numerical instability) and a large number of training iterations. In the next sections we will implement these conclusions in practice.

## 4   Matching weight increment statistics for different mini-batch sizes

First, to correct the different diffusion rates observed for different batch sizes, we will aim to match the statistics of the weights increments to that of a small batch size.

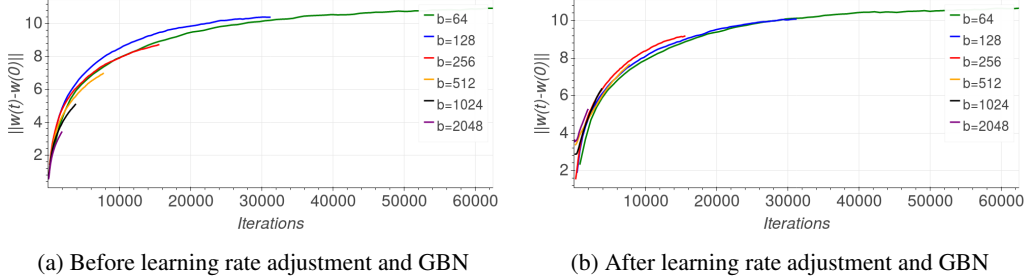

(a) Before learning rate adjustment and GBN      (b) After learning rate adjustment and GBN

Figure 2: Euclidean distance of weight vector from initialization

**Learning rate.** Recall that in this paper we investigate SGD, possibly with momentum, where the weight updates are proportional to the estimated gradient.

$$\Delta \mathbf{w} \propto \eta \hat{\mathbf{g}}, \tag{5}$$

where $\eta$ is the learning rate, and we ignore for now the effect of batch normalization.

In appendix section A, we show that the covariance matrix of the parameters update step $\Delta \mathbf{w}$ is,

$$\text{cov}\left(\Delta \mathbf{w}, \Delta \mathbf{w}\right) \approx \frac{\eta^2}{M} \left( \frac{1}{N} \sum_{n=1}^{N} \mathbf{g}_n \mathbf{g}_n^\top \right) \tag{6}$$

in the case of uniform sampling of the mini-batch indices (with or without replacement), when $M \ll N$. Therefore, a simple way to make sure that the covariance matrix stays the same for all mini-batch sizes is to choose

$$\eta \propto \sqrt{M}, \tag{7}$$

i.e., we should increase the learning rate by the square root of the mini-batch size.

We note that Krizhevsky (2014) suggested a similar learning rate scaling in order to keep the variance in the gradient expectation constant, but chose to use a linear scaling heuristics as it reached better empirical result in his setting. Later on, Li (2017) suggested the same.

Naturally, such an increase in the learning rate also increases the mean steps $\mathbb{E}\left[\Delta \mathbf{w}\right]$. However, we found that this effect is negligible since $\mathbb{E}\left[\Delta \mathbf{w}\right]$ is typically orders of magnitude lower than the standard deviation.

Furthermore, we can match both the first and second order statistics by adding multiplicative noise to the gradient estimate as follows:

$$\hat{\mathbf{g}} = \frac{1}{M} \sum_{n \in B}^{N} \mathbf{g}_n z_n,$$

where $z_n \sim \mathcal{N}\left(1, \sigma^2\right)$ are independent random Gaussian variables for which $\sigma^2 \propto M$. This can be verified by using similar calculation as in appendix section A. This method keeps the covariance constant when we change the batch size, yet does not change the mean steps $\mathbb{E}\left[\Delta \mathbf{w}\right]$.

In both cases, for the first few iterations, we had to clip or normalize the gradients to prevent divergence. Since both methods yielded similar performance [5] (due the negligible effect of the first order statistics), we preferred to use the simpler learning rate method.

It is important to note that other types of noise (e.g., dropout (Srivastava et al., 2014), dropconnect (Wan et al., 2013), label noise (Szegedy et al., 2016)) change the structure of the covariance matrix and not just its scale, thus the second order statistics of the small batch increment cannot be accurately matched. Accordingly, we did not find that these types of noise helped to reduce the generalization gap for large batch sizes.

Lastly, note that in our discussion above (and the derivations provided in appendix section A) we assumed each per-sample gradient $\mathbf{g}_n$ does not depend on the selected mini-batch. However, this ignores the influence of batch normalization. We take this into consideration in the next subsection.

**Ghost Batch Normalization.** Batch Normalization (BN) (Ioffe & Szegedy, 2015), is known to accelerate the training, increase the robustness of neural network to different initialization schemes and improve generalization. Nonetheless, since BN uses the batch statistics it is bounded to depend on the choosen batch size. We study this dependency and observe that by acquiring the statistics on small virtual ("ghost") batches instead of the real large batch we can reduce the generalization error. In our experiments we found out that it is important to use the full batch statistic as suggested by (Ioffe & Szegedy, 2015) for the inference phase. Full details are given in Algorithm 1. This modification by itself reduce the generalization error substantially.

---

**Algorithm 1:** Ghost Batch Normalization (GBN), applied to activation $x$ over a large batch $B_L$ with virtual mini-batch $B_S$. Where $B_S < B_L$.

---

**Require:** Values of $x$ over a large-batch: $B_L = \{x_{1...m}\}$ size of virtual batch $|B_S|$; Parameters to be learned: $\gamma$, $\beta$, momentum $\eta$

**Training Phase:**

Scatter $B_L$ to $\{X^1, X^2, ...X^{|B_L|/|B_S|}\} = \{x_{1...|B_S|}, x_{|B_S|+1...2|B_S|} \cdots x_{|B_L|-|B_S|...m}\}$

$\mu_B^l \leftarrow \frac{1}{|B_S|}\sum_{i=1}^{|B_S|} X_i^l$ for $l = 1,2,3...$ {calculate ghost mini-batches means}

$\sigma_B^l \leftarrow \sqrt{\frac{1}{|B_S|}\sum_{i=1}^{|B_S|}(X_i^l - \mu_B)^2 + \epsilon}$ for $l = 1,2,3...$ {calculate ghost mini-batches std}

$\mu_{run} = (1-\eta)^{|B_S|}\mu_{run} + \sum_{i=1}^{|B_L|/|B_S|}(1-\eta)^i \cdot \eta \cdot \mu_B^l$

$\sigma_{run} = (1-\eta)^{|B_S|}\sigma_{run} + \sum_{i=1}^{|B_L|/|B_S|}(1-\eta)^i \cdot \eta \cdot \sigma_B^l$

**return** $\gamma\frac{X^l - \mu_B^l}{\sigma_B^l} + \beta$

**Test Phase:**

**return** $\gamma\frac{X - \mu_{run}^l}{\sigma_{run}} + \beta$ {scale and shift}

---

We note that in a multi-device distributed setting, some of the benefits of "Ghost BN" may already occur, since batch-normalization is often preformed on each device separately to avoid additional communication cost. Thus, each device computes the batch norm statistics using only its samples (i.e., part of the whole mini-batch). It is a known fact, yet unpublished, to the best of the authors knowledge, that this form of batch norm update helps generalization and yields better results than computing the batch-norm statistics over the entire batch. Note that GBN enables flexibility in the small (virtual) batch size which is not provided by the commercial frameworks (e.g., TensorFlow, PyTorch) in which the batch statistics is calculated on the entire, per-device, batch. Moreover, in those commercial frameworks, the running statistics are usually computed differently from "Ghost BN", by weighting each update part equally. In our experiments we found it to worsen the generalization performance.

Implementing both the learning rate and GBN adjustments seem to improve generalization performance, as we shall see in section 6. Additionally, as can be seen in Figure 6, the slopes of the logarithmic weight distance graphs seem to better matched, indicating similar diffusion rates. We also observe some constant shift, which we believe is related to the gradient clipping. Since this shift only increased the weight distances, we assume it does not harm the performance.

## 5 Adapting number of weight updates eliminates generalization gap

According to our conclusions in section 3, the initial high-learning rate training phase enables the model to reach farther locations in the parameter space, which may be necessary to find wider local minima and better generalization. Examining figure 2b, the next obvious step to match the graphs for different batch sizes is to increase the number of training iterations in the initial high learning rate regime. And indeed we noticed that the distance between the current weight and the initialization point can be a good measure to decide upon when to decrease the learning rate.

Note that this is different from common practices. Usually, practitioners decrease the learning rate after validation error appears to reach a plateau. This practice is due to the long-held belief that the optimization process should not be allowed to decrease the training error when validation error "flatlines", for fear of overfitting (Girosi et al., 1995). However, we observed that substantial improvement to the final accuracy can be obtained by continuing the optimization using the same

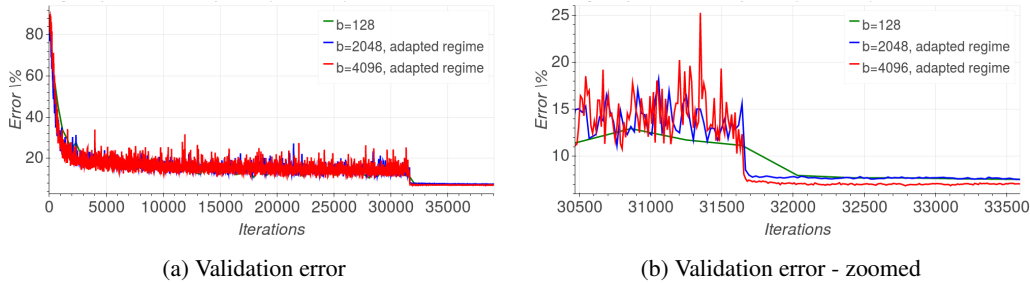

| (a) Validation error | (b) Validation error - zoomed |

Figure 3: Comparing generalization of large-batch regimes, adapted to match performance of small-batch training.

learning rate even if the training error decreases while the validation plateaus. Subsequent learning rate drops resulted with a sharp validation error decrease, and better generalization for the final model.

These observations led us to believe that "generalization gap" phenomenon stems from the relatively small number of updates rather than the batch size. Specifically, using the insights from Figure 2 and our model, we adapted the training regime to better suit the usage of large mini-batch. We "stretched" the time-frame of the optimization process, where each time period of $e$ epochs in the original regime, will be transformed to $\frac{|B_L|}{|B_S|}e$ epochs according to the mini-batch size used. This modification ensures that the number of optimization steps taken is identical to those performed in the small batch regime. As can be seen in Figure 3, combining this modification with learning rate adjustment completely eliminates the generalization gap observed earlier [6].

# 6 Experiments

**Experimental setting.** We experimented with a set of popular image classification tasks:

- MNIST (LeCun et al., 1998b) - Consists of a training set of $60K$ and a test set of $10K$ $28 \times 28$ gray-scale images representing digits ranging from 0 to 9.
- CIFAR-10 and CIFAR-100 (Krizhevsky, 2009) - Each consists of a training set of size 50K and a test set of size $10K$. Instance are $32 \times 32$ color images representing 10 or 100 classes.
- ImageNet classification task Deng et al. (2009) - Consists of a training set of size 1.2M samples and test set of size 50K. Each instance is labeled with one of 1000 categories.

To validate our findings, we used a representative choice of neural network models. We used the fully-connected model, F1, as well as shallow convolutional models C1 and C3 suggested by Keskar et al. (2017). As a demonstration of more current architectures, we used the models: VGG (Simonyan, 2014) and Resnet44 (He et al., 2016) for CIFAR10 dataset, Wide-Resnet16-4 (Zagoruyko, 2016) for CIFAR100 dataset and Alexnet (Krizhevsky, 2014) for ImageNet dataset.

In each of the experiments, we used the training regime suggested by the original work, together with a momentum SGD optimizer. We use a batch of 4096 samples as "large batch" (LB) and a small batch (SB) of either 128 (F1,C1,VGG,Resnet44,C3,Alexnet) or 256 (WResnet). We compare the original training baseline for small and large batch, as well as the following methods[7]:

- Learning rate tuning (LB+LR): Using a large batch, while adapting the learning rate to be larger so that $\eta_L = \sqrt{\frac{|B_L|}{|B_S|}}\eta_S$ where $\eta_S$ is the original learning rate used for small batch, $\eta_L$ is the adapted learning rate and $|B_L|, |B_S|$ are the large and small batch sizes, respectively.
- Ghost batch norm (LB+LR+GBN): Additionally using the "Ghost batch normalization" method in our training procedure. The "ghost batch size" used is 128.
- Regime adaptation: Using the tuned learning rate as well as ghost batch-norm, but with an adapted training regime. The training regime is modified to have the same number of

iterations for each batch size used - effectively multiplying the number of epochs by the relative size of the large batch.

**Results.** Following our experiments, we can establish an empirical basis to our claims. Observing the final validation accuracy displayed in Table 1, we can see that in accordance with previous works the move from a small-batch (SB) to a large-batch (LB) indeed incurs a substantial generalization gap. However, modifying the learning-rate used for large-batch (+LR) causes much of this gap to diminish, following with an additional improvement by using the Ghost-BN method (+GBN). Finally, we can see that the generalization gap completely disappears when the training regime is adapted (+RA), yielding validation accuracy that is good-as or better than the one obtained using a small batch.

We additionally display results obtained on the more challenging ImageNet dataset in Table 2 which shows similar impact for our methods.

Table 1: Validation accuracy results, SB/LB represent small and large batch respectively. GBN stands for Ghost-BN, and RA stands for regime adaptation

| Network | Dataset | SB | LB | +LR | +GBN | +RA |
|---|---|---|---|---|---|---|
| F1 (Keskar et al., 2017) | MNIST | 98.27% | 97.05% | 97.55% | 97.60% | 98.53% |
| C1 (Keskar et al., 2017) | Cifar10 | 87.80% | 83.95% | 86.15% | 86.4% | 88.20% |
| Resnet44 (He et al., 2016) | Cifar10 | 92.83% | 86.10% | 89.30% | 90.50% | 93.07% |
| VGG (Simonyan, 2014) | Cifar10 | 92.30% | 84.1% | 88.6% | 91.50% | 93.03% |
| C3 (Keskar et al., 2017) | Cifar100 | 61.25% | 51.50% | 57.38% | 57.5% | 63.20% |
| WResnet16-4 (Zagoruyko, 2016) | Cifar100 | 73.70% | 68.15% | 69.05% | 71.20% | 73.57% |

Table 2: ImageNet top-1 results using Alexnet topology (Krizhevsky, 2014), notation as in Table 1.

| Network | LB size | Dataset | SB | LB[8] | +LR[8] | +GBN | +RA |
|---|---|---|---|---|---|---|---|
| Alexnet | 4096 | ImageNet | 57.10% | 41.23% | 53.25% | 54.92% | 59.5% |
| Alexnet | 8192 | ImageNet | 57.10% | 41.23% | 53.25% | 53.93% | 59.5% |

## 7 Discussion

There are two important issues regarding the use of large batch sizes. First, why do we get worse generalization with a larger batch, and how do we avoid this behaviour? Second, can we decrease the training wall clock time by using a larger batch (exploiting parallelization), while retaining the same generalization performance as in small batch?

This work tackles the first issue by investigating the random walk behaviour of SGD and the relationship of its diffusion rate to the size of a batch. Based on this and empirical observations, we propose simple set of remedies to close down the generalization gap between the small and large batch training strategies: (1) Use SGD with momentum, gradient clipping, and a decreasing learning rate schedule; (2) adapt the learning rate with batch size (we used a square root scaling); (3) compute batch-norm statistics over several partitions ("ghost batch-norm"); and (4) use a sufficient number of high learning rate training iterations.

Thus, the main point arising from our results is that, in contrast to previous conception, there is no inherent generalization problem with training using large mini batches. That is, model training using large mini-batches can generalize as well as models trained using small mini-batches. Though this answers the first issues, the second issue remained open: can we speed up training by using large batch sizes?

Not long after our paper first appeared, this issue was also answered. Using a Resnet model on Imagenet Goyal et al. (2017) showed that, indeed, significant speedups in training could be achieved using a large batch size. This further highlights the ideas brought in this work and their importance to future scale-up, especially since Goyal et al. (2017) used similar training practices to those we

described above. The main difference between our works is the use of a linear scaling of the learning rate[9], similarly to Krizhevsky (2014), and as suggested by Bottou (2010). However, we found that linear scaling works less well on CIFAR10, and later work found that linear scaling rules work less well for other architectures on ImageNet (You et al., 2017).

We also note that current "rules of thumb" regarding optimization regime and explicitly learning rate annealing schedule may be misguided. We showed that good generalization can result from extensive amount of gradient updates in which there is no apparent validation error change and training error continues to drop, in contrast to common practice. After our work appeared, Soudry et al. (2017) suggested an explanation to this, and to the logarithmic increase in the weight distance observed in Figure 2. We show this behavior happens even in simple logistic regression problems with separable data. In this case, we exactly solve the asymptotic dynamics and prove that $\mathbf{w}(t) = \log(t)\hat{\mathbf{w}} + O(1)$ where $\hat{\mathbf{w}}$ is to the $L_2$ maximum margin separator. Therefore, the margin (affecting generalization) improves slowly (as $O(1/\log(t))$), even while the training error is very low. Future work, based on this, may be focused on finding when and how the learning rate should be decreased while training.

**Conclusion.**  In this work we make a first attempt to tackle the "generalization gap" phenomenon. We argue that the initial learning phase can be described using a high-dimensional "random walk on a random potential" process, with a an "ultra-slow" logarithmic increase in the distance of the weights from their initialization, as we observe empirically. Following this observation we suggest several techniques which enable training with large batch without suffering from performance degradation. This implies that the problem is not related to the batch size but rather to the amount of updates. Moreover we introduce a simple yet efficient algorithm "Ghost-BN" which improves the generalization performance significantly while keeping the training time intact.

### Acknowledgments

We wish to thank Nir Ailon, Dar Gilboa, Kfir Levy and Igor Berman for their feedback on the initial manuscript. The research was partially supported by the Taub Foundation, and the Intelligence Advanced Research Projects Activity (IARPA) via Department of Interior/ Interior Business Center (DoI/IBC) contract number D16PC00003. The U.S. Government is authorized to reproduce and distribute reprints for Governmental purposes notwithstanding any copyright annotation thereon. Disclaimer: The views and conclusions contained herein are those of the authors and should not be interpreted as necessarily representing the official policies or endorsements, either expressed or implied, of IARPA, DoI/IBC, or the U.S. Government.

## Footnotes

[2]It was later pointed out (Dinh et al., 2017) that certain "degenerate" directions, in which the parameters can be changed without affecting the loss, must be excluded from this explanation. For example, for any $c > 0$ and any neuron, we can multiply all input weights by $c$ and divide the output weights by $c$: this does not affect the loss, but can generate arbitrarily strong positive curvature.

[3]Either exactly (with replacement) or approximately (without replacement): see appendix section A.

[4]Note that this form is consistent with eq. (1), if $f(x) = x^{\alpha/2}$.

[5] a simple comparison can be seen in appendix (figure 3)

[6]Additional graphs, including comparison to non-adapted regime, are available in appendix (figure 2).

[7]Code is available at https://github.com/eladhoffer/bigBatch.

[8] Due to memory limitation those experiments were conducted with batch of 2048.

[9]e.g., Goyal et al. (2017) also used an initial warm-phase for the learning rate, however, this has a similar effect to the gradient clipping we used, since this clipping was mostly active during the initial steps of training.

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
