[Supplementary Material · appendix.pdf]

# Supplementary Material for "Train longer, generalize better: closing the generalization gap in large batch training regime of neural networks"

# Appendix

## A   Derivation of eq. (6)

Note that we can write the mini-batch gradient as

$$\hat{\mathbf{g}} = \frac{1}{M} \sum_{n=1}^{N} \mathbf{g}_n s_n \text{ with } s_n \triangleq \begin{cases} 1 & , \text{ if } n \in B \\ 0 & , \text{ if } n \notin B \end{cases}$$

Clearly, $\hat{\mathbf{g}}$ is an unbiased estimator of $\mathbf{g}$, if

$$\mathbb{E} s_n = P(s_n = 1) = \frac{M}{N}.$$

since then

$$\mathbb{E}\hat{\mathbf{g}} = \frac{1}{M} \sum_{n=1}^{N} \mathbf{g}_n \mathbb{E} s_n = \frac{1}{N} \sum_{n=1}^{N} \mathbf{g}_n = \mathbf{g}.$$

First, we consider the simpler case of sampling with replacement. In this case it easy to see that different minibatches are uncorrelated, and we have

$$\mathbb{E}[s_n s_{n'}] = P(s_n = 1)\,\delta_{nn'} + P(s_n = 1, s_{n'} = 1)(1 - \delta_{nn'})$$
$$= \frac{M}{N}\delta_{nn'} + \frac{M^2}{N^2}(1 - \delta_{nn'}).$$

and therefore

$$\begin{aligned}
\text{cov}(\hat{\mathbf{g}}, \hat{\mathbf{g}}) &= \mathbb{E}\left[\hat{\mathbf{g}}\hat{\mathbf{g}}^\top\right] - \mathbb{E}\hat{\mathbf{g}}\,\mathbb{E}\hat{\mathbf{g}}^\top \\
&= \frac{1}{M^2} \sum_{n=1}^{N} \sum_{n'=1}^{N} \mathbb{E}[s_n s_{n'}]\,\mathbf{g}_n \mathbf{g}_{n'}^\top - \mathbf{g}\mathbf{g}^\top \\
&= \frac{1}{M^2} \sum_{n=1}^{N} \sum_{n'=1}^{N} \left[\frac{M}{N}\delta_{nn'} + \frac{M^2}{N^2}(1 - \delta_{nn'})\right] \mathbf{g}_n \mathbf{g}_{n'}^\top - \mathbf{g}\mathbf{g}^\top \\
&= \left(\frac{1}{M} - \frac{1}{N}\right)\frac{1}{N} \sum_{n=1}^{N} \mathbf{g}_n \mathbf{g}_n^\top,
\end{aligned}$$

which confirms eq. (6).

Next, we consider the case of sampling without replacement. In this case the selector variables are now different and correlated between different mini-batches (*e.g.*, with indices $t$ and $t+k$), since we

cannot select previous samples. Thus, these variables $s_n^t$ and $s_n^{t+k}$ have the following second-order statistics

$$\mathbb{E}\left[s_n^t s_{n'}^{t+k}\right] = P\left(s_n^t = 1, s_{n'}^{t+k} = 1\right)$$
$$= \frac{M}{N}\delta_{nn'}\delta_{k0} + \frac{M}{N}\frac{M}{N-1}\left(1 - \delta_{nn'}\delta_{k0}\right).$$

This implies

$$\mathbb{E}\left[\hat{\mathbf{g}}_t\hat{\mathbf{g}}_{t+k}^\top\right] - \mathbb{E}\hat{\mathbf{g}}_t\mathbb{E}\hat{\mathbf{g}}_{t+k}^\top$$
$$= \mathbb{E}\left[\left(\frac{1}{M}\sum_{n=1}^N s_n^t\mathbf{g}_n\right)\left(\frac{1}{M}\sum_{n'=1}^N s_{n'}^{t+k}\mathbf{g}_{n'}^\top\right)\right] - \mathbf{g}\mathbf{g}^\top$$
$$= \frac{1}{M^2}\sum_{n=1}^N\sum_{n'=1}^N \mathbb{E}\left[s_n^t s_{n'}^{t+k}\right]\mathbf{g}_n\mathbf{g}_{n'}^\top - \mathbf{g}\mathbf{g}^\top$$
$$= \frac{1}{M^2}\sum_{n=1}^N\sum_{n'=1}^N\left[\left(\frac{M}{N} - \frac{M^2}{N^2-N}\right)\delta_{nn'}\delta_{k0} - \frac{M^2}{N^2-N}\right]\mathbf{g}_n\mathbf{g}_{n'}^\top - \mathbf{g}\mathbf{g}^\top$$

so, if $k=0$ the covariance is

$$\mathbb{E}\left[\hat{\mathbf{g}}_t\hat{\mathbf{g}}_t^\top\right] - \mathbb{E}\hat{\mathbf{g}}_t\mathbb{E}\hat{\mathbf{g}}_t^\top = \left(\frac{1}{M} - \frac{1}{N-1}\right)\frac{1}{N}\sum_{n=1}^N\mathbf{g}_n\mathbf{g}_n^\top + \frac{1}{N-1}\mathbf{g}\mathbf{g}^\top$$
$$\overset{M\ll N}{\approx} \frac{1}{M}\left(\frac{1}{N}\sum_{n=1}^N\mathbf{g}_n\mathbf{g}_n^\top\right)$$

while the covariance between different minibatches ($k \neq 0$) is much smaller for $M \ll N$

$$\mathbb{E}\left[\hat{\mathbf{g}}_t\hat{\mathbf{g}}_{t+k}^\top\right] - \mathbb{E}\hat{\mathbf{g}}_t\mathbb{E}\hat{\mathbf{g}}_{t+k}^\top = \frac{1}{N-1}\mathbf{g}\mathbf{g}^\top$$

this again confirms eq. (6).

## B Estimating $\alpha$ from random potential

The logarithmic increase in weight distance (Figure 2 in the paper) matches a "random walk on a random potential" model with $\alpha = 2$. In such a model the loss auto-covariance asymptotically increases with the square of the weight distance, or, equivalently (Marinari et al., 1983), the standard deviation of the loss difference asymptotically increases linearly with the weight distance

$$\text{std} \triangleq \sqrt{\mathbb{E}\left(L\left(\mathbf{w}\right) - L\left(\mathbf{w}_0\right)\right)^2} \sim \|\mathbf{w} - \mathbf{w}_0\|. \tag{1}$$

In this section we examine this behavior: in Figure we indeed find such a linear behavior, confirming the prediction of our model with $\alpha = 2$.

To obtain the relevant statistics to plot eq. 1 we conducted the following experiment on Resnet44 model (He et al., 2016). We initialized the model weights, $\mathbf{w}_0$, according to Glorot & Bengio (2010), and repeated the following steps a 1000 times, given some parameter $c$:

- Sample a random direction $\mathbf{v}$ with norm one.
- Sample a scalar $z$ uniformly in some range $[0, c]$.
- Choose $\mathbf{w} = \mathbf{w}_0 + z\mathbf{v}$.
- Save $\|\mathbf{w} - \mathbf{w}_0\|$ and $L(\mathbf{w})$.

We have set the parameter $c$ so that the maximum weight distance from initialization $\|\mathbf{w} - \mathbf{w}_0\|$ is equal to the same maximal distance in Figure 2 in the paper, *i.e.*, $c \approx 10$.

Figure 1: **The standard deviation of the loss shows linear dependence on weight distance (eq. 1) as predicted by the "random walk on a random potential" model with $\alpha = 2$ we found in the main paper.** To approximate the ensemble average in eq. 1 we divided the x-axis to $b$ bins and calculated the empiric average in each bin. Each panel shows the resulting graph for a different value of $b$.

Figure 2: Comparing regime adapted large batch training vs. a 2048 batch with no adaptation.

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

Figure 3: Comparing a learning scale fix for a 2048 batch, to a multiplicative noise to the gradient of the same scale

Figure 4: Comparing $L_2$ distance from initial weight for different batch sizes