[Reviews · NeurIPS 2017]

Reviewer 1



I think the paper provides some clarity on a topic that has seen a bit of attention lately, namely that of the role of noise in optimization and in particular the hypothesis of sharp minima/flat minima. From this perspective I think this datapoint is important for our collective understanding of training deep networks. I don’t think the observation made by the authors come as a surprise to anyone with experience with these models, however the final conclusion might. I.e. we know the progress in learning is proportional to the number of updates (and not amount of examples seen). We know that when having large minibatches we have lower variance and hence we should use a larger learning rate, etc. I think one practical issue that people have got stuck in the past is that with larger minibatches the computational cost of any given gradient increases. Nowadays this is mediated to some extend by parallelization though maybe not completely. However, in this light, because of the additional computational cost, one would hope for there to be a gain, and hence one to get away with fewer steps (i.e. such that clock wall time is lower) otherwise why bother. I think this practical consideration might stopped people to reach the natural conclusion the authors have. Some comments: 1. I don’t fully follow the use of sharp/flat minima used by the authors. For example they say that you need more steps to reach a flat minima. I’m not sure about this statement. I think the crucial observation is that you might need more steps (because the distance you travel is proportional with log number of steps) to reach a minima. I’m not sure the diffusion process says anything about sharp vs flat. I would rather think this view is more natural to think in terms of saddle local minima. And to reach a saddle with a lower index one needs to take more steps. I think this marginally affects the message of the paper, but I think strictly phrasing this paper in terms of Keskar et al might not be the best choices. 2. Regarding my comment with respect to wall-clock time. If using larger minibatches we need to correct such that we do the same number of updates, what is the gain (or reason) for pushing for large minibatches in practice? I understand the theoretical argument (that helps understanding generalization in neural nets). But does this imply also that at most we might have a similar time cost as we increase the minibatch size? Can we do something about it? Maybe this is a more important question rather than figuring out how to parallelize large minibatches, which is the message I took from lines 270-271. 3. Ghost Batch Normalization. Is my understanding that this is mostly an empirical finding? What is the justification for one to require noisy estimates of mean and variance? It feels like a failing of Batch Norm rather than SGD (and I assume that SGD without batch norm reaches same performance with small batches and large batches by just playing with learning rate and number of updates). I feel that maybe a better understanding of this would be useful. 4. Line 91, did you mean \hat{g} instead of g_n? 5. Any news on the ImageNet results with everything?

Reviewer 2



Summary of the paper and some comments: This paper investigates the reasons for generalization gap, namely the detrimental effect of large mini-batches on the generalization performance of the neural networks. The paper argues that the reason why the generalization performance is worse when the minibatch size is smaller is due to the less number of steps that the model is trained for. They also discuss the relationship between their method and recent works on flat-minima for generalization. The main message of this paper is that there is no fundamental issue with SGD using large minibatches, and it is possible to make SGD + large minibatches to generalize well as well. The authors propose three different methods to overcome these issues, 1) Ghost BN: They have proposed to use smaller minibatch sizes during the training to compute the population statistics such as mean and the standard deviation. During the test time, they use the full-batch statistics. I think computing the population statistics over the small minibatches, probably have some sort of regularization effect coming from the noise in the estimates. I feel like this is not very well justified in the paper. 2) Learning rate tuning, they propose to rescale the learning rate of the large mini-batches proportionally to the size of the minibatch. This intuitively makes sense, but authors should discuss more and justify it in a better way. 3) Regime adaptation, namely trying to use the same number of iterations as the model being trained. This is justified in the first few sections of the paper. Firstly, the paper at some points is very verbose and feel like uses the valuable space to explain very simple concepts, e.g. the beginning of section 3 and lacks to provide enough justifications for their approach. The paper discusses mainly the generalization of SGD with large minibatches, but there is also an aspect of convergence of SGD as well. Because the convergence results of SGD also depends on the minibatch size as well, please see [1]. I would like to see a discussion of this as well. In my opinion, anyone that spend on monitoring the learning process of the neural networks would notice easily notice that the learning curves usually have a logarithmic behavior. The relationship to the random-walk theory is interesting, but the relationship that is established in this theory is very weak, it is only obtained by approximating an approximation. Equation 4 is interesting, for instance, if we initialize w_0 to be all zero, it implies that the norm of the w_t will grow logarithmically with respect to the number of iterations. This is also an interesting property, in my opinion. Experiments are very difficult to interpret, in particular when those plots are printed on the paper. The notation is a bit inconsistent in throughout the paper, some parts of the paper use bracket for the expectation and some other parts don't. Authors should explicitly state what the expectation is taken over. In Figure 3, can you also consider adding the large minibatch cases without regime adaptation for comparison? It would be interesting to see the learning curves of the models with respect to different wallclock time as well. I can not see the AB in Table 1 and 2. The results and the improvements are marginal, sometimes it hurts. It seems like largest improvement comes from RA. The paper could be better written. I think the paragraph sections in the abstract are unnecessary. The section 7 could be renamed as "Discussion and Conclusion" and could be simplified. [1] Li, Mu, et al. "Efficient mini-batch training for stochastic optimization." Proceedings of the 20th ACM SIGKDD international conference on Knowledge discovery and data mining. ACM, 2014.

Reviewer 3



The paper addresses the recently observed generalization gap problem between large and small batch methods. It has been hypothesized that methods with small and large batches converge to very different optima. This work relates the process of gradient descent with an ultra slow diffusion process and show that this gap can be explained by lower diffusion time for the batch method. Based on this observation a correction is devised that makes the diffusion curves match for different batch sizes. Furthermore, this could help explain why training long past the time where the validation error has plateaued is necessary and still improves performance. This work is quite interesting and will stimulate further work into understanding certain strange properties of SGD. The experimental section contains experiments on MNIST, CIFAR-10 and 100 as well as Imagenet. They compare against the results of (Keskar et al, 2016) and show that their tricks allow the large batch methods to generalize better than the baseline. One key finding is that the models should be trained much longer with large batches - which in a sense negates their purpose. What is not clear from the experiments is how necessary Ghost-Batch normalization is. Would good results be obtained with regime adaptation only? This would strengthen the story of the paper a bit more. the Section 4 proposes a multiplicative noise gradient but this is not addressed in the experimental section. It would have been interesting to see some results on this. Even just in Figure 2.